# A New Graphical Method for Displaying Two-Dimensional Echocardiography Results in Dogs: Comprehensive Analysis of Results of Diagnostic Imaging Organized in a BOX (CARDIOBOX)

**DOI:** 10.3390/vetsci12010034

**Published:** 2025-01-09

**Authors:** Federico J. Curra-Gagliano, Martín Ceballos, José I. Redondo, Javier Engel-Manchado

**Affiliations:** 1Cátedra de Medicina 1, Facultad de Ciencias Veterinarias, Universidad de Buenos Aires, Buenos Aires C1427CWO, Argentina; 2Cátedra Anestesiología y Algiología, Facultad de Ciencias Veterinarias, Universidad de Buenos Aires, Buenos Aires C1427CWO, Argentina; 3Departamento de Medicina y Cirugía Animal, Facultad de Veterinaria, Universidad Cardenal Herrera-CEU, CEU Universities, 46115 Valencia, Spain; nacho@uchceu.es; 4Internal Medicine, Veterinary Medicine and Therapeutic Research Group, Faculty of Veterinary Science, Research Institute of Biomedical and Health Sciences (IUIBS), University of Las Palmas de Gran Canaria, 35413 Las Palmas de Gran Canaria, Spain; jengelmanchado@gmail.com; 5HeartBeatVet, 46025 Valencia, Spain

**Keywords:** echocardiography, dogs, graphic method, graphic result interpretation, veterinary cardiology, clinical decision-making

## Abstract

Rapid and accurate interpretation of echocardiographic results in dogs is crucial to good clinical decision-making. In this study, we developed and evaluated a graphical method called CARDIOBOX, designed to simplify the visual representation of echocardiographic values. CARDIOBOX is a system consisting of three overlapping boxes, which house nine boxes. Each box represents a specific range of echocardiographic values. Its design is based on the results of an analysis of 2967 dogs. In this process, percentiles and cut-off points were established to delimit the boxes corresponding to normal and abnormal ranges. To evaluate the efficacy of CARDIOBOX, 55 veterinarians participated in the interpretation of echocardiographs of dogs, presented in two different formats: using CARDIOBOX and exclusively using numerical reference tables. The time taken to interpret and the accuracy of responses in each format were recorded. The results indicated that CARDIOBOX enabled veterinarians to interpret echocardiographic values more quickly without decreasing interpretation efficiency. This study shows that CARDIOBOX allows veterinarians to interpret and apply echocardiographic results more quickly and effectively, significantly facilitating clinical decision-making.

## 1. Introduction

The exact prevalence of cardiac disease in dogs is unknown, although it may be around 10% of cases seen in daily clinical practice [1]. Echocardiography is the technique of choice in diagnosing and staging most cardiac diseases [2,3]. Echocardiographic data are usually interpreted employing tables, which use allometric formulas to predict average values based on a dog’s weight [4].

Interpreting numerical values can be challenging for the human brain, designed to interpret graphs more effectively than numbers [5,6,7]. Human medicine has developed new strategies to present data innovatively and impactfully [8]. The efficient design of new graphical representation systems requires fast, simple, and effective implementation and interpretation, and they have been successfully employed in clinical pathology [9].

Thus, a new way of displaying these complex echocardiographic measurements could be through their graphic representation, normalising the values beforehand so that they can be easily interpreted independently of the animal’s weight, as in the allometric tables used in veterinary cardiology [4]. Given the challenges faced by patients of varying sizes and weights in a clinical setting, this study sought to create, standardise, explain, and assess a visual system for interpreting echocardiographic findings, called CARDIOBOX (Comprehensive Analysis of Results of Diagnostic Imaging Organized in a BOX).

The objectives of this study were as follows: (a) to design a method for graphical representation of standardised echocardiographic results and (b) to validate the method’s usefulness in a clinical setting. We hypothesise that CARDIOBOX allows the representation of echocardiographic numerical results in a simple graph that is easy and efficient to interpret and that its interpretation is faster than the traditional numerical method.

## 2. Material and Methods

This project received authorisation from the Institutional Committee for the Care and Use of Experimental Animals (CICUAL) at the Faculty of Veterinary Sciences, University of Buenos Aires, Argentina (Project 2019/49). All animal owners were informed about this study’s objectives, methods, and purpose and gave their informed consent, which was also signed by the veterinarians involved in this clinical study. This study comprised two phases: (1) the design and standardisation of the CARDIOBOX method using echocardiographic values in dogs as a reference and (2) assessing the method’s effectiveness in a clinical context.

### 2.1. Design and Standardisation of the CARDIOBOX Method

This descriptive, cross-sectional study was conducted to create and standardise the CARDIOBOX graphical method for assessing dog echocardiographic results. This study included 2967 dogs.

#### 2.1.1. Patient Selection, Assessment, and Classification

All animals underwent a comprehensive evaluation, including anamnesis, detailed physical examination, and complementary diagnostic tests. These tests involved measuring systolic, diastolic, and mean arterial blood pressure with a SunTech Vet 30 oscillometric monitor (SunTech Medical Inc., Morrisville, USA), performing a six-channel electrocardiogram using a TEMIS model TM-300 digital electrocardiograph (Temis Tech., Cordoba, Argentina), right lateral and ventrodorsal radiographic views, and haematology including complete blood count (red blood cells, white blood cells, and platelets, as well as haemoglobin and haematocrit). Biochemistry, including glucose, urea, creatinine, alkaline phosphate (ALKP), cholesterol, alanine aminotransferase (ALT), total protein, globulin, albumin, and electrolytes (sodium, potassium, and chloride), was also measured.

Based on these assessments, animals were assigned to one of two groups according to health status. The HEALTHY GROUP (802 dogs) comprised dogs with no cardiac or systemic disease signs, excluding breeds with specific echocardiographic reference values (e.g., greyhounds, Labradors). Dogs diagnosed with cardiac disease formed the CARDIAC GROUP (2165 dogs). Together, the HEALTHY GROUP and the CARDIAC GROUP were called the TOTAL GROUP (2967 dogs).

#### 2.1.2. Echocardiographic Evaluation

Echocardiography was performed following standard clinical recommendations [10] using a SonoScape A6 ultrasound machine (SonoScape Medical Corp., Guangdong, China) with 5–9 MHz microconvex probes and a Vinno 5 ultrasound machine (Vinno, Suzhou City, China) using 1–5 MHz adult phased array and 5–11 MHz microconvex probes. The left atrium and ascending aorta diameters were measured in B-mode in the right parasternal short-axis view at early diastole to calculate the left atrium/aorta (LA/Ao) index [11]. Additionally, in the right parasternal short-axis M-mode view at the level of the papillary muscles, the following parameters were measured:Interventricular septal thickness in diastole and systole (IVSd and IVSs)Left ventricular internal diameter in diastole and systole (LVIDd and LVIDs)Left ventricular free-wall thickness in diastole and systole (PLVWd and PLVWs)The principal investigator (FCG) performed all echocardiographic scans, and one additional investigator (JEM), each with over ten years of experience and specialised accreditation in veterinary cardiology, reviewed the images. Images that did not meet quality standards for accurate measurements were excluded from the analysis. Echocardiographic values for each patient were standardised to account for body weight variations using a formula derived from Cornell et al. (2004) [4]:

Standardised Value = Absolute Value (cm)/Weight (kg) ^B^, where B is the constant *b* in the allometric equation defined by Cornell et al. (2004) [4].

#### 2.1.3. Construction of the CARDIOBOX Graphical Method

The CARDIOBOX is a visual tool developed to evaluate graphically echocardiographic parameters in dogs. Each parameter is represented within a row of nine boxes, divided into three categories: values below the normal range (boxes 1, 2, and 3), values within the normal range (central grey boxes 4, 5, and 6), and values above the normal range (boxes 7, 8, and 9). Each box represents a specific percentile range, allowing an intuitive interpretation of a patient’s data against reference values derived from a healthy population (see Figure 1).

The boundaries for each box in a specific parameter were determined as follows:

Define overall bounds (minimum and maximum): The minimum and maximum values were established using data from the TOTAL POPULATION, including HEALTHY and CARDIAC groups (*n* = 2967). The minimum value defines the lower bound of box 1, while the maximum value defines the upper bound of box 9.Calculate boxes 4, 5, and 6 (normal range): The central normal range was established using data from the HEALTHY GROUP (*n* = 802). This standard range spans from the 1st to the 99th percentiles and consists of three central boxes. The intervals between boxes 4, 5, and 6 are equal, with box five centred precisely at the median:
○Box 5 is centred on the median (50th percentile), representing the midpoint of the normal distribution.○Box 4 begins at the 1st percentile of the data of the HEALTHY GROUP and extends to a central boundary between the 1st percentile and the median, defining the transition from box 4 to box 5.○Box 6 begins at this equidistant boundary on the other side of the median and extends to the 99th percentile of the data of the HEALTHY GROUP.
Calculate boxes 1, 2, 3 and 7, 8, 9 (values outside the normal range): To represent values below and above the normal range, the following steps were applied:○Boxes 1, 2, and 3 capture values below the 1st percentile of the normal range (the lower bound of box 4). These boxes are set with equidistant boundaries that span the range from the minimum value (the lower bound of box 1) to the start of the normal range.○Boxes 7, 8, and 9 capture values above the 99th percentile of the normal range (the upper bound of box 6). These boxes are also set with equidistant boundaries, covering the range from the end of the normal range to the maximum value (the upper bound of box 9).

### 2.2. Validation of the CARDIOBOX

A survey was designed to assess the clinical utility of the CARDIOBOX graphical tool. This survey was created following the CHERRIES checklist [12]. The survey was developed and hosted on the JOTFORM platform (https://form.jotform.com/223278319852059, last accessed on 19 August 2023, also accessible on Appendix A), automatically recording each question’s response time. The survey was active from 25 November to 6 December 2022. It was distributed in specialised veterinary groups, including cardiology, anaesthesiology, and general clinical practice. Participation was anonymous and voluntary.

The survey included four prominent clinical cases with echocardiographic findings: two patients without pathology and two with myxomatous mitral valve disease (MMVD) at different stages. Each case was presented using two formats: (a) a panel of numerical echocardiographic values accompanied by a reference table and (b) a CARDIOBOX chart displaying the values graphically.

The four cases were presented randomly, two using the CARDIOBOX method (one healthy and one pathological case) and two using the traditional numerical method with reference values (one healthy and one pathological case).

Key variables studied included response time (automatically measured by JOTFORM) and the percentage of correct answers for the seven variables under analysis. Additionally, participants were asked to rate the effectiveness and ease of use of the CARDIOBOX method compared to the traditional numerical format using a 5-point Likert scale. A final question gauged their willingness to adopt CARDIOBOX in clinical practice.

#### Statistical Analysis

The statistical analysis was done using the R language program (4.4.2). The first step is calculating the percentiles for the studied standardised echocardiographic variables to build the CARDIOBOX graph.

In the second step, a detailed statistical analysis was conducted to assess the effect of the presentation method (CARDIOBOX vs. numerical) and the presence of pathology on response time and accuracy. It was also used to analyse participants’ perceptions using Likert-type questions.

The parameters analysed in the survey were response time (TIME) and percentage of correct answers (CORRECT), which were compared in two cases: CARDIOBOX and cases with numerical results and reference tables. Using the Shapiro–Wilk test, the null hypothesis of normality could be rejected. As the response data (time and correct answers) did not have a normal distribution, to validate the results of the form and detect statistical differences, the Wilcoxon test was used to compare the variables’ percentage of correct answers and response time between groups. The significance level was set at 5% (*p* < 0.05).

Descriptive statistics, such as means and standard deviations, were calculated for response time and percentage of correct responses, grouped by method of presentation and presence of pathology. Shapiro–Wilk normality tests indicated that the data did not follow a normal distribution, justifying non-parametric tests in subsequent analyses.

The non-parametric Friedman test for repeated-measures data was applied to compare the median response time and percentage of correct responses among the different cases. After finding significant differences, post hoc comparisons were performed using the Wilcoxon paired signed-rank test with Bonferroni correction to adjust the significance level due to multiple comparisons. A linear mixed model was also used to analyse the effect of the method of presentation, presence of pathology, and order of presentation on response time and percentage of correct responses. These models allow for repeated measures of data and intra- and inter-subject variability. The lmer function of the lme4 R package was used, incorporating fixed effects for the method of assessment, the presence or absence of pathology, and the order in which cases were presented, as well as their interactions, in addition to a random effect for the participant’s identifier (identifier variable). A similar model was fitted for the percentage of correct answers.

Cronbach’s alpha coefficient was calculated to measure the reliability of the scales used in the questionnaire and assess the internal consistency of the Likert-type questions related to the perceived ease of use and effectiveness of CARDIOBOX.

## 3. Results

A total of 2967 dogs participated in this study. Of these, 802 were classified as healthy, forming the HEALTHY GROUP, while 2165 were diagnosed with cardiac diseases and were included in the CARDIAC GROUP. Table 1 shows the calculated percentiles for data parameters for the HEALTHY GROUP and TOTAL GROUP. The CARDIOBOX boxes’ limits were determined and shown in Table 2 using this data.

### Validation of the CARDIOBOX

A total of 55 surveys were collected. Results indicated that evaluation times were significantly shorter with CARDIOBOX than with the traditional numerical method (*p* < 0.0001), with no significant time difference between cases with or without pathology (CARDIOBOX: *p* = 0.28; Numeric: *p* = 0.80). CARDIOBOX also yielded a higher percentage of correct answers compared to the numerical method (*p* = 0.02793), and there was no significant difference in accuracy between cases with and without pathology as shown in Table 3 (CARDIOBOX: *p* = 1; Numeric: *p* = 0.224).

Participants’ perceptions of the CARDIOBOX graphical tool were also assessed using Likert-type questions, and the effect of the presentation method (traditional vs. CARDIOBOX) was analysed.

Mean scores and standard deviations were obtained for two critical aspects of the tool. Ease of use received a mean score of 4.49 (out of 5) with a standard deviation of 0.98, indicating that participants perceived the tool as easy to use. The tool’s effectiveness received a mean score of 4.18 (out of 5) with a standard deviation of 1.19, suggesting that users consider CARDIOBOX effective for interpreting echocardiographic results.

The frequency analysis showed that most participants gave high scores in terms of ease of use: 40 people rated it a five, 7 a four, and only 8 gave scores of three or lower. For the tool’s effectiveness, 32 participants rated it a five and 10 a four, while 13 scored three or less. Results are shown in Figure 2.

Interest in the future implementation of CARDIOBOX was high, with 44 respondents (80%) considering it beneficial for patient information. A smaller group of 10 respondents (18.18%) expressed uncertainty about its usefulness, while only 1 (1.82%) deemed it irrelevant.

Cronbach’s alpha was calculated for the first two Likert-type questions (ease of use and effectiveness of the tool), obtaining a value of 0.76. This result indicates good internal consistency between items, supporting the measures’ reliability.

For the analysis of turnaround time and accuracy, a linear mixed model was fitted to assess the effect of presentation method, presence of pathology, and order of presentation on turnaround time. The results showed that the traditional method was associated with a significant increase in turnaround time compared to CARDIOBOX. Specifically, using the conventional method increased response time by 49.49 s (standard error = 4.65, *t*-value = 10.64). Pathology had no significant effect on response time (*t*-value = −0.18), nor did pathology significantly affect the order of presentation (*t*-value = 1.21).

A linear mixed model was used to calculate the percentage of correct answers. The results indicated that the traditional method tended to slightly decrease the rate of correct answers compared to CARDIOBOX, with a reduction of 0.0299 (standard error = 0.0154, *t*-value = −1.94). However, this effect did not reach statistical significance. The presence of pathology showed no significant impact on the accuracy of responses (*t*-value = 1.47), and the order of presentation had no considerable effect either (*t*-value = 1.27). Analysis of the impact of the order of presentation revealed that this factor had no significant influence on either response time or the percentage of correct answers. This indicates that no learning or fatigue effects were observed that could bias this study’s results.

The Likert-type questions indicated a positive trend in participants’ perceptions of CARDIOBOX. Most participants gave high ratings for ease of use and effectiveness, and many showed interest in its future implementation.

## 4. Discussion

The CARDIOBOX graphical method was developed to provide a standardised, efficient, and intuitive means for interpreting dog echocardiographic parameters. Our findings support our hypothesis that CARDIOBOX could facilitate rapid and reliable clinical assessments. Clinicians found the tool user-friendly and effective in distinguishing between healthy and pathological cases, with significantly faster assessment times than traditional numerical methods, such as the Cornell allometric scaling charts [4].

Defining the range limits within the nine CARDIOBOX boxes was critical for clinical accuracy and utility. The method was constructed using a percentile-based approach, where the outer bounds (minimum and maximum) were defined from the TOTAL POPULATION, which included both healthy and cardiac groups. At the same time, the central boxes (4, 5, and 6) were determined exclusively from the HEALTHY GROUP. This combined approach aims to balance sensitivity for pathological cases with specificity for healthy cases, providing a reliable diagnostic aid for veterinarians.

Using the TOTAL POPULATION to define the minimum and maximum values (boxes 1 and 9) ensures that CARDIOBOX captures the full range of values encountered in practice, enhancing sensitivity by encompassing the entire data spectrum. This is essential, as a narrow outer range might exclude clinically relevant values. Including data from both healthy and cardiac groups at the extremes allows CARDIOBOX to reflect the broad echocardiographic variation seen in practice, making it suitable for diverse cases [4,13]. However, defining the central boxes (4, 5, and 6) from the HEALTHY GROUP ensures that standard range cutoffs remain strictly derived from non-pathological values, preserving high specificity. Previous research suggests that using only healthy populations for reference intervals is crucial for avoiding skewed ranges, which could misrepresent the normal distribution [3].

Another potential approach to constructing the CARDIOBOX would be to define the central and extreme boxes exclusively from the HEALTHY GROUP data. While this might yield a tighter distribution, such a configuration would likely result in pathological values clustering at the outermost edges (boxes 1 and 9). This clustering could reduce diagnostic efficacy, as it may limit the method’s ability to differentiate degrees of abnormality, making distinguishing between mild and severe pathology harder. Practical diagnostic tools benefit from capturing a spectrum of abnormal values, enabling clinicians to detect deviations at varying stages [8]. By using healthy-only data to set the normal range while relying on the broader total population for the extremes, CARDIOBOX enhances both specificity and sensitivity across clinical cases.

CARDIOBOX was validated by 55 veterinarians, who responded positively to its ease of use and effectiveness. While the feedback suggests high user satisfaction, expanding this study to include veterinarians with varying experience in cardiology would provide more comprehensive validation. Additional insights from general practitioners and specialists would help confirm that CARDIOBOX is universally accessible. Given that veterinary technology can sometimes face resistance due to learning curves, the ease-of-use scores indicate that this tool could integrate smoothly into clinical practice, regardless of users’ prior experience [13].

A further consideration is this study’s focus on dogs under 20 kg, which restricts generalisability to larger breeds with differing cardiac morphologies. The variability in cardiac parameters across breeds underscores the importance of expanding CARDIOBOX to include larger dogs and diverse morphologies and validating its applicability for other canine populations. Further research in this direction would strengthen CARDIOBOX as a versatile tool and extend its clinical relevance.

Standardised echocardiographic values adjusted for weight through allometric scaling ensure consistency in measurement across dogs of different body sizes (see Appendix A). However, a more detailed explanation of how allometric scaling integrates with the graphical method could provide a more robust theoretical basis. This explanation would enhance readers’ understanding of the method’s applicability across different body sizes, further supporting CARDIOBOX’s validity as a tool for diverse veterinary populations.

Although this study provides valuable cross-sectional data, longitudinal studies could offer additional insights into how well CARDIOBOX performs over time, especially in monitoring disease progression or assessing treatment efficacy. Applying CARDIOBOX in a longitudinal context could reveal its potential for tracking individual changes in echocardiographic parameters, broadening its utility in veterinary practice.

In human medicine, graphical data representation is widely recognised as beneficial for diagnostic accuracy, interpretive speed, and error reduction, mainly when it balances central tendencies with extremes [14]. CARDIOBOX follows these principles by adopting percentile-based limits, enhancing sensitivity and specificity for a reliable, clinically useful tool. By employing a clear, percentile-based visual differentiation, CARDIOBOX supports prompt and accurate decision-making in clinical settings [15]. Additionally, user feedback in this study highlights the positive reception of CARDIOBOX’s layout, suggesting that graphical tools of this type could be beneficial across veterinary and human medicine for complex data interpretation [13].

The unique advantages of this new graphical method are summarized as follows: (A) It solves a critical problem. Traditional numerical representation of echocardiographic data often leads to delays and errors in interpretation, particularly in scenarios requiring quick decisions or for practitioners without immediate access to reference materials (see Appendix A). By providing a straightforward graphical representation, the method significantly reduces interpretation time while maintaining diagnostic accuracy. Furthermore, (B) it fills an existing gap. While echocardiography and its result-representation methods are well-established, no simple and easily adaptable method exists for future integration into ultrasound software. This approach bridges the gap by enabling automated, reliable, and simplified interpretations directly within ultrasound devices, empowering clinicians to make immediate decisions without external resources. Lastly, (C) it simplifies and modernizes an existing solution. Beyond its graphical design, it is built to seamlessly integrate with emerging technologies such as Deep Learning and Artificial Intelligence (AI). These advancements promise faster and more accurate interpretations, while the system’s scalability allows practitioners to build personalized reference tables tailored to their patients and clinical approaches, making it a dynamic and evolving tool for modern veterinary practice. It has the potential to be integrated into ultrasound software, facilitating real-time graphical representation and automatic interpretation of echocardiographic data, which further enhances its utility in clinical settings. CARDIOBOX could integrate AI and machine learning to enhance diagnostic capabilities. Developing automated algorithms for segmentation and measurement could provide clinicians with immediate, standardized echocardiographic assessments. Tests in simulated environments show that using computer vision in ultrasound devices can streamline data acquisition and offer real-time feedback in CARDIOBOX outputs. These improvements would aid in monitoring disease progression and treatment efficacy, expanding CARDIOBOX’s impact in veterinary cardiology.

This study has certain limitations. At present, this first study does not consider all the echocardiographic parameters. In this publication, we have focused on introducing the CARDIOBOX system using the main absolute parameters that vary with body weight. In addition, the CARDIOBOX system is validated on breeds with standard weight-adjusted reference tables, excluding Greyhounds, Labradors, and other giant breeds that are not represented in the demographic studied, as well as specific brachycephalic breeds that require customised reference tables. Future research should investigate the applicability of this system to these excluded breeds, also exploring variations between breed types (brachycephalic, mesocephalic, and dolichocephalic) and size (small vs. giant). In addition, extending the research to other species, such as cats and even humans, is essential to realise the full potential of this system. In conclusion, CARDIOBOX represents an effective tool for interpreting echocardiographic results in dogs, significantly improving interpretation speed without compromising diagnostic accuracy. The balance between sensitivity and specificity through the combined use of healthy and total population data enhances its applicability across clinical scenarios, from general practice to specialised cardiology. However, expanding the tool’s validation to larger breeds, and conducting longitudinal studies would further strengthen CARDIOBOX’s potential. These findings suggest that CARDIOBOX could become a widely applicable alternative to traditional methods, advancing clinical data interpretation and optimising patient care in veterinary medicine.

## Figures and Tables

**Figure 1 vetsci-12-00034-f001:**
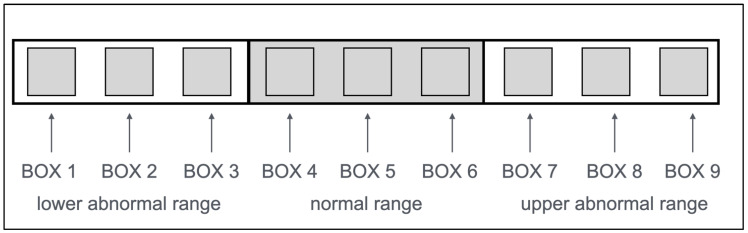
Basic structure of the CARDIOBOX. A marker indicates each parameter in one of the nine boxes: boxes 1, 2, and 3 for values below the normal range (white); boxes 4, 5, and 6 for values within the normal range (grey central rectangle); and boxes 7, 8, and 9 for values above the normal range (white).

**Figure 2 vetsci-12-00034-f002:**
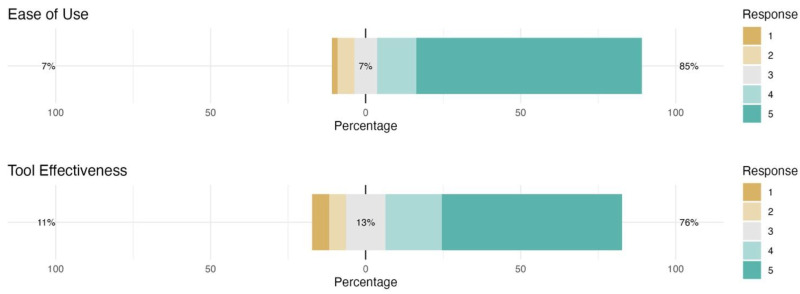
Likert graphs for perceived ease of use and tool effectiveness.

**Table 1 vetsci-12-00034-t001:** Calculated percentiles using the normalised data from 802 healthy dogs and the total population (*n* = 2967).

	Percentile										
HEALTHY GROUP	Min	1	2.5	5	25	50	75	95	97.5	99	Max
AIDn	0.49	0.56	0.59	0.62	0.70	0.75	0.82	0.92	0.96	1.00	1.23
IVSSDn	0.25	0.29	0.31	0.32	0.38	0.42	0.46	0.54	0.56	0.58	0.69
LVIDDn	0.84	1.12	1.19	1.22	1.38	1.47	1.55	1.66	1.68	1.69	1.74
PLVWDn	0.23	0.26	0.29	0.30	0.37	0.42	0.47	0.56	0.58	0.62	0.69
IVSSn	0.33	0.43	0.43	0.46	0.54	0.59	0.64	0.74	0.77	0.82	0.92
LVIDSn	0.28	0.47	0.54	0.58	0.73	0.82	0.90	1.01	1.04	1.06	1.09
PLVWSn	0.37	0.41	0.44	0.47	0.54	0.61	0.68	0.77	0.80	0.83	0.95
TOTAL POPULATION											
AIDn	0.35	0.58	0.61	0.64	0.73	0.80	0.91	1.20	1.37	1.52	2.22
IVSSDn	0.22	0.28	0.30	0.32	0.38	0.43	0.48	0.56	0.60	0.65	0.84
LVIDDn	0.68	1.07	1.16	1.22	1.41	1.54	1.69	2.12	2.29	2.46	3.04
PLVWDn	0.19	0.27	0.29	0.31	0.38	0.43	0.49	0.57	0.61	0.65	0.86
IVSSn	0.11	0.40	0.44	0.47	0.55	0.62	0.68	0.81	0.85	0.90	1.22
LVIDSn	0.24	0.45	0.52	0.59	0.75	0.86	0.97	1.18	1.34	1.58	2.55
PLVWSn	0.32	0.41	0.44	0.47	0.56	0.63	0.70	0.81	0.85	0.89	1.12

AIDn: normalised left atrial internal diameter in diastole. IVSSDn: normalised interventricular septal thickness in diastole. LVIDDn: normalised left ventricular internal diameter at diastole. PLVWDn: normalised left ventricular free-wall thickness in diastole. IVSSn: normalised interventricular septal thickness in systole. LVIDSn: normalised left ventricular internal diameter at systole. PLVWSn: normalised left ventricular free-wall thickness in systole.

**Table 2 vetsci-12-00034-t002:** Limits for each box for constructing the CARDIOBOX for the studied echocardiographic variables.

	BOX 1		BOX 2		BOX 3		BOX 4		BOX 5		BOX 6		BOX 7		BOX 8		BOX 9	
AIDn	0.35	0.41	0.42	0.48	0.49	0.55	0.56	0.70	0.71	0.85	0.86	1.00	1.01	1.26	1.27	1.52	1.53	2.22
IVSSDn	0.22	0.24	0.25	0.26	0.27	0.28	0.29	0.39	0.40	0.48	0.49	0.58	0.59	0.61	0.62	0.65	0.66	0.84
LVIDDn	0.68	0.82	0.83	0.96	0.97	1.11	1.12	1.31	1.32	1.50	1.51	1.69	1.70	2.07	2.08	2.46	2.47	3.04
PLVWDn	0.19	0.21	0.22	0.23	0.24	0.25	0.26	0.38	0.39	0.50	0.51	0.62	0.63	0.63	0.64	0.65	0.66	0.86
IVSSn	0.11	0.21	0.22	0.32	0.33	0.42	0.43	0.56	0.57	0.69	0.70	0.82	0.83	0.85	0.86	0.90	0.91	1.22
LVIDSn	0.24	0.31	0.32	0.39	0.40	0.46	0.47	0.67	0.68	0.86	0.87	1.06	1.07	1.32	1.33	1.58	1.59	2.55
PLVWSn	0.32	0.35	0.36	0.38	0.39	0.40	0.41	0.55	0.56	0.69	0.70	0.83	0.84	0.86	0.87	0.89	0.90	1.12

AIDn: normalised left atrial internal diameter in diastole. IVSSDn: normalised interventricular septal thickness in diastole. LVIDDn: normalised left ventricular internal diameter at diastole. PLVWDn: normalised left ventricular free-wall thickness in diastole. IVSSn: normalised interventricular septal thickness in systole. LVIDSn: normalised left ventricular internal diameter at systole. PLVWSn: normalised left ventricular free-wall thickness in systole.

**Table 3 vetsci-12-00034-t003:** Mean time and standard deviation employed in the evaluation of echocardiographic data from the cases using the traditional numerical method and the proposed graphical method (CARDIOBOX) and percentage of correct answers in evaluating echocardiographic data of the cases using the conventional numeric method and the proposed graphical method (CARDIOBOX).

Method	Pathology	Mean Time	SD Time	Correct
CARDIOBOX	NO	65	35	97.7%
CARDIOBOX	YES	72	32	94.6%
Numeric	NO	109	52	92.7%
Numeric	YES	128	61	93.5%

## Data Availability

Data supporting the reported results can be sent to anyone interested by contacting the corresponding author.

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
