# Peer review of "A New Graphical Method for Displaying Two-Dimensional Echocardiography Results in Dogs: Comprehensive Analysis of Results of Diagnostic Imaging Organized in a BOX (CARDIOBOX)"

_vetsci, 2025, doi:10.3390/vetsci12010034_

Round 1
Reviewer 1 Report
Comments and Suggestions for Authors
I have read and reviewed this manuscript with great interest and overall, from this reviewer's perspective, it is an experimental study that has been well-planned and executed. Overall it is a study with refreshingly simple wording that is easy to understand. Other strengths of the manuscript that I can highlight are the following: the introduction provides sufficient background and includes pertinent references, the research design is adequate, and the methods are repeatable and correctly described. The conclusions are supported by the results.
Nevertheless, some points must be addressed to achieve publication quality. I have left some comments hoping that they can help the authors.
General comments
L97-98: Please indicate the radiographic projections used and the analytes or parameters measured in blood cell count, blood chemistry, and electrolytes.
L104: please indicate the sample size of each study group.
L117: Did CARDIOBOX be included in the evaluation; ejection fraction, shortening fraction, and EPSS? Please clarify.
L394: I suggest to the authors that they discuss limitations and even perspectives concerning variables such as racial type (brachycephalic, mesocephalic, dolichocephalic), the size of the breed (small—giant), the application of the scale to patients undergoing treatment due to heart disease, and, where appropriate, the prospect of use in other species, such as cats.
Author Response
Dear Editor,
We have carefully assessed the comments and suggestions provided by the reviewers and would like to express our gratitude for your encouraging words about our work. Your positive feedback reinforces our confidence in the significance of CARDIOBOX as an innovative tool for veterinary cardiology. We are delighted you recognise its potential to streamline clinical workflows and enhance decision-making processes. We are confident that the revisions will further improve the quality and impact of the article.
We now turn to the comments of both reviewers.
The authors.
Reviewer 1
We sincerely thank you for your detailed and thoughtful comments, which have greatly helped us improve the clarity and quality of our manuscript. Your insightful feedback has been instrumental in refining the manuscript, and we sincerely appreciate the time and effort you invested in reviewing our work.
General comments:
L97-98: Please indicate the radiographic projections used and the analytes or parameters measured in blood cell count, blood chemistry, and electrolytes.
In the text (lines 98-102), we have added your suggestion: the radiographic views are right lateral and ventrodorsal, while haematology includes complete blood count (red blood cells, white blood cells, and platelets. Also, haemoglobin and haematocrit). Biochemistry included glucose, urea, creatinine, alkaline phosphate (ALKP), cholesterol, alanine aminotransferase (ALT), total protein, globulin, albumin, and electrolytes (sodium, potassium and chloride).
L104: please indicate the sample size of each study group.
Thank you for your comment. This point has been clarified later in the paper, but we agree that reviewing the number of patients in each group is beneficial. We have updated the information in lines 103-108 as it stands follows:
“Based on these assessments, animals were assigned to one of two groups according to health status. The HEALTHY GROUP (802 dogs) comprised dogs with no cardiac or systemic disease signs, excluding breeds with specific echocardiographic reference values (e.g., greyhounds, Labradors). Dogs diagnosed with cardiac disease formed the CARDIAC GROUP (2165 dogs). Together, the HEALTHY GROUP and the CARDIAC GROUP were called the TOTAL GROUP (2967 dogs).”
L117: Was CARDIOBOX included in the evaluation, including ejection fraction, shortening fraction, and EPSS? Please clarify.
Yes, thank you for pointing this out. The CARDIOBOX project encompasses additional parameters; however, in this initial paper, we have focused mainly on introducing the CARDIOBOX SYSTEM using the core absolute parameters that vary with body weight. In subsequent publications, we plan to expand on this by including these additional parameters: ejection fraction, shortening fraction, EPSS, E wave, A wave, etc., and extending the analysis to cover various breeds and species. We will include this in the limitations, as you can read in the following answer.
L394: I suggest that the authors discuss limitations and even perspectives concerning variables such as racial type (brachycephalic, mesocephalic, dolichocephalic), breed size (small—giant), the application of the scale to patients undergoing treatment for heart disease, and where appropriate, the prospect of use in other species, such as cats.
Thank you for highlighting this specific observation. We have included this limitations paragraph at the end of the conclusions, lines 406-416:
“This study has certain limitations. At present, this first study does not consider all the echocardiographic parameters. In this publication, we have focused on introducing the CARDIOBOX system using the main absolute parameters that vary with body weight. In addition, the CARDIOBOX system is validated on breeds with standard weight-adjusted reference tables, excluding Greyhounds, Labradors, and other giant breeds that are not represented in the demographic studied, as well as specific brachycephalic breeds that require customised reference tables. Future research should investigate the applicability of this system to these excluded breeds, also exploring variations between breed types (brachycephalic, mesocephalic, and dolichocephalic) and size (small vs giant). In addition, extending the research to other species, such as cats and even humans, is essential to realise the full potential of this system.”
Reviewer 2 Report
Comments and Suggestions for Authors
This study introduces and validates CARDIOBOX, an innovative graphical tool designed to present echocardiographic data in dogs. Developed using a comprehensive dataset of 2,967 dogs (802 classified as healthy and 2,165 diagnosed with cardiac conditions), the system employs percentile-based ranges to categorize echocardiographic values into nine distinct boxes. CARDIOBOX demonstrated significantly faster interpretation times compared to traditional numerical methods, without any compromise in accuracy. Its simplicity and effectiveness have been positively received by veterinarians, underscoring its potential to enhance clinical decision-making in the field of veterinary cardiology.
Novelty
This study represents a significant advancement in veterinary diagnostic imaging by introducing a graphical approach to interpreting echocardiographic parameters. Unlike traditional numerical tables, CARDIOBOX provides an intuitive visual representation that enables faster and more accurate data interpretation. This method aligns with established trends in human medical imaging, where graphical tools have proven to improve both efficiency and accuracy. As such, CARDIOBOX stands out as a valuable contribution to veterinary cardiology, with the potential to streamline clinical workflows and elevate decision-making processes.
Minor changes:
-Addressing the tool's limitations regarding breed diversity and weight variation would enhance its impact. Future studies should explore its utility in larger breeds or specific breeds with unique echocardiographic profiles (e.g., greyhounds, Labradors).
-While the manuscript mentions integration with emerging AI technologies, a clearer roadmap or examples of such integration would add value. Consider including preliminary data or a framework for integrating CARDIOBOX into diagnostic devices.
- The text contains several repetitions that, if removed, could improve the manuscript's flow. An example of this can be found in line 80.
Line 80: "Provided informed consent, and veterinarians participating in the clinical survey signed a consent form."
Author Response
Reviewer 2
We sincerely thank you for your thorough and insightful comments, which have significantly enhanced the quality of our manuscript. Your valuable feedback has helped us refine our arguments and expand on key aspects, particularly the potential integration of CARDIOBOX with emerging AI technologies. We deeply appreciate their dedicated time and expertise in reviewing our work.
Addressing the tool's limitations regarding breed diversity and weight variation would enhance its impact. Future studies should explore its utility in more extensive or specific breeds with unique echocardiographic profiles (e.g., greyhounds, and Labradors).
Thank you for pointing out this precise observation. We decided to include at the end of conclusions, line 406-416, this limitations paragraph:
“This study has certain limitations. At present, this first study does not consider all the echocardiographic parameters. In this publication, we have focused on introducing the CARDIOBOX system using the main absolute parameters that vary with body weight. In addition, the CARDIOBOX system is validated on breeds with standard weight-adjusted reference tables, excluding Greyhounds, Labradors, and other giant breeds that are not represented in the demographic studied, as well as specific brachycephalic breeds that require customised reference tables. Future research should investigate the applicability of this system to these excluded breeds, also exploring variations between breed types (brachycephalic, mesocephalic, and dolichocephalic) and size (small vs giant). In addition, extending the research to other species, such as cats and even humans, is essential to realise the full potential of this system.”
While the manuscript mentions integration with emerging AI technologies, a more precise roadmap or examples would add value. Consider including preliminary data or a framework for integrating CARDIOBOX into diagnostic devices.
We appreciate the reviewer’s suggestion. We agree that this is a promising direction for enhancing diagnostic efficiency. In future work, we plan to use machine learning and computer vision to automate cardiac structure segmentation and echocardiographic measurement, enabling a more streamlined workflow. Automated algorithms could assign measurement results to the relevant CARDIOBOX category, reducing interpretation time and operator dependency. Furthermore, integrating these algorithms into diagnostic devices like network-connected ultrasound machines may generate real-time CARDIOBOX visualisations, which is advantageous for longitudinal cardiac monitoring conditions. We included a paragraph in the discussion section pointing out your suggestion, lines 398-404.
“CARDIOBOX could integrate AI and machine learning to enhance diagnostic capabilities. Developing automated algorithms for segmentation and measurement could provide clinicians with immediate, standardized echocardiographic assessments. Tests in simulated environments show that using computer vision in ultrasound devices can streamline data acquisition and offer real-time feedback in CARDIOBOX outputs. These improvements would aid in monitoring disease progression and treatment efficacy, expanding CARDIOBOX's impact in veterinary medicine cardiology”.
The text contains several repetitions that, if removed, could improve the manuscript's flow. An example of this can be found in line 80.
Line 80: "Provided informed consent, and veterinarians participating in the clinical survey signed a consent form."
Thank you for pointing this out; we decided to improve that line after this review, as you can read in lines 80-82:
“All animal owners were informed about the study's objectives, methods and purpose and gave their informed consent, which was also signed by the veterinarians involved in the clinical study.”